# The Clean Your Plate Campaign: Resisting Table Food Waste in an Unstable World

**Lingfei Wang** , **Yuqin Yang and Guoyan Wang ***

School of Communication, Dushu Lake Campus of Soochow University, Suzhou 215006, China;
20204042004@stu.suda.edu.cn (L.W.); 20204242016@stu.suda.edu.cn (Y.Y.)
* Correspondence: gywang@suda.edu.cn; Tel./Fax: +86-512-6588-1608

**Abstract:** The COVID-19 pandemic threatens global food security and has created an urgent need for food conservation. This article presents a review of clean plate campaigns around the world. It aims to fight food waste and reveal the factors that may influence food waste. The Clean Plate Club in the US developed during wartime and relied heavily on political power for compliance, whereas the Clean Plate movement in South Korea was based on religion. China's Clean Your Plate Campaign (CYPC) has gone through two stages: CYPC I and CYPC II. The latter occurred during the unstable period of the COVID-19 pandemic. It was large-scale and more strongly enforced than CYPC I. In China, CYPC has relied more on personal virtue than on politics or religion. Culture, policy, COVID-19, and behavior are all important social factors that influence food waste. Specifically, two cultural values are drivers of food waste in China: hospitality and face-saving (*mianzi*). In terms of policy, China's food waste law mainly relies on persuasion; it lacks any power of enforcement. Laws in France and Italy, by contrast, focus on re-using food and involve both coercion and incentives. COVID-19 may have led to panic purchasing and stockpiling, but, in general, it has resulted in a reduction in food waste.

**Keywords:** clean your plate campaigns; food waste; consumption stage; food policy; China

## 1. Introduction

Food conservation is an issue of global importance. In unstable conditions such as pandemics and wars, clean plate campaigns have been developed to limit food waste around the world. The COVID-19 pandemic has caused unprecedented disruptions to food systems, leading to food shortages and food waste across the supply chain [1]. Currently, nearly 690 million people globally go hungry and the COVID-19 pandemic may add between 83 and 132 million people to that number [2]. By the middle of this century, the world's population is projected to reach 9.1 billion, and as a result, food production must increase by about 70 percent [3]. In the context of increasingly scarce and degraded natural resources, reducing food waste will be integral to increasing food availability in the future [4]. However, food waste has always been a serious issue. Roughly one-third of the food produced globally for human consumption is lost or wasted [5]. The latest report estimated that food waste at the consumer level (household and food service) appears to be more than twice the 2011 FAO estimate [6]. Moreover, food waste not only threatens global food security but also consumes substantial global resources and causes serious environmental pollution and economic losses [7,8]. Reducing food waste is thus an urgent and critical issue.

Food waste occurs when food that is appropriate for human consumption is discarded [9]. It occurs at the stages of consumption and retail. Food loss, meanwhile, occurs at all points along the food supply chain from harvest up to, but not including, retail [3]. A review found that the consumption stage represents a major factor of increasing importance to food waste [10]: consumers are the biggest contributor to the total volume of

food waste generated [11,12]. It is usually believed that in developing countries, food loss occurs mainly during the earlier stages of the food supply chain. In developed countries, by contrast, food waste is more likely to occur at the consumer stage [13,14]. The latest report published by UNEP found that action on food waste is equally important in high-, upper-middle-, and lower-middle-income countries. This diverges from earlier narratives that have mainly concentrated on consumer food waste in developed countries [6]. Emerging economies, particularly China, are likely to play a critical role in determining global food waste at mid-century [6,15].

In response to the COVID-19 pandemic, on 11 August 2020, Chinese President Xi Jinping issued guidance on reducing food waste. The Clean Your Plate Campaign (CYPC, sometimes also referred to as the Clear Your Plate Campaign) was introduced. In April 2021, China's first official anti-food waste law was enacted, encouraging the promotion of the CYPC. The slogan of the CYPC is "Start with me, no leftovers today". It encourages people to eat all the food on their plates. Researchers have studied similar clean plate campaigns in the US [16,17] and South Korea [18,19], but relatively few studies have engaged in international comparisons of clean plate campaigns. In this review paper, we present a comparative study of these campaigns. This paper summarizes the possible factors, such as culture and policy, that may influence food waste. The research questions addressed are:

RQ1: What kinds of clean plate campaigns have been carried out?
RQ2: What are the factors that influence table food waste?
RQ3: In terms of policy factors, what are the characteristics and effects of the various anti-food waste laws?

## 2. Clean Plate Campaigns around the World
### 2.1. Basic Description

In China, the Clean Your Plate Campaign was officially launched in January 2013 by the nonprofit organization IN_33. It was promoted both online and offline (CYPC I). In response to an article on CYPC published by the Xinhua News Agency, President Xi Jinping made a written comment that food waste must be eliminated, which gave a significant boost to the movement. Mirosa et al. [20] found that many of the posts regarding CYPC on Sina Weibo appeared after Xi Jinping's statements. On 11 August 2020, Xi Jinping once again issued guidance about reducing food waste. The New Clean Your Plate Campaign (CYPC II) was thus launched [20,21]. On 29 April 2021, China's first anti-food waste law was enacted, encouraging the CYPC.

Other countries have similar clean plate campaigns. In 1917, the United States Congress launched the Clean Plate Club to combat food shortages during World War I. Herbert Hoover, America's first Food Administrator, was the head of this campaign and proclaimed that "Food will win the war". He appealed to US adults' sense of patriotism to encourage them to change their eating habits. At school, children were required to recite a pledge that read, "At table I'll not leave a scrap of food upon my plate. And I'll not eat between meals, but for supper time I'll wait." The US Food Administration was disbanded immediately after the war. However, in the aftermath of World War II (1947), Harry S. Truman resurrected Hoover's campaign in order to deal with another bout of food scarcity. In this iteration, it was presented to elementary school children as a club.

South Korea also started a Clean Plate movement in 2004, which was initiated by a Buddhist Non-Governmental organization (NGO) named EcoBuddha. About 1000 volunteers of EcoBuddha, most of whom were Buddhist housewives, played a key role. Before the movement, they practiced Buddhist asceticism and were already trained to eat in a clean plate manner in their normal life. EcoBuddha's approach to the Clean Plate movement involves awakening to the interrelationship between human beings and nature, based on a series of Buddhist lectures and the practice of asceticism [18].

### 2.2. Comparison of Different Countries

CYPC II occurred during the COVID-19 pandemic, an unstable period. It occurred on a large scale and was more strongly enforced than CYPC I. Shortly before CYPC I, the central government issued "Eight Regulations" (4 December 2012), which aimed to put an end to officials' extravagant feasts and receptions. In this way, the government supported the implementation of CYPC I [22,23]. For reasons of national security during the COVID-19 pandemic, the government launched CYPC II [22]. It was later accompanied by relevant regulations and laws. During this stage, the government targeted the public as well as officials [20]. The groups who implemented this campaign ranged from a single social organization to all levels of government from top to bottom.

As can be seen from Table 1, the American Clean Plate Club and CYPC II in China were both rooted in a specific period and driven by the government. However, there are several differences between them. First, the need for food conservation and the degree of urgency for conservation vary. During the two world wars, food shortages could have led to a national demise, but this was not the case during COVID-19. According to experts, China's food security was not seriously affected by COVID-19 [24]. Second, the incentives for the public to perform are different. Winning or losing a war is directly related to personal survival, and war can trigger strong patriotic passions. However, in China, the anti-food waste law did not introduce penalties for food waste by the general public. The current regulations are aimed at the government, universities, caterers, and other institutions. Generally, food conservation is now a matter of interest and obligation for all Americans, whereas it is still only a voluntary act for most Chinese people.

**Table 1.** Similar clean plate campaigns.

|  | United States | South Korea | China | |
|---|---|---|---|---|
| **Name** | Clean Plate Club | Clean Plate movement | CYPC I | CYPC II |
| **Time period** | Wartime | Daily | Daily | COVID-19 |
| **Originator** | Government agency | Formal social organization | Informal social organization | Government |
| **Scale** | 500,000 women volunteers, fourteen million families supported | 1000 volunteers, 1.5 million people pledged, pledge fund of 140,000 dollars was gathered | Nearly 30 members, CYPC was reposted 50 million times on Weibo, 60,000 leaflets were distributed, and over 5000 posters were put up in Beijing | Nationwide, conducted at all levels of government from top to bottom, with full participation of the catering industry and schools |
| **The role of the state** | Proposer, leader | Supporter, collaborators | Supporter | Proposer, leader |
| **Main tactics** | Political power, patriotic fervor | Religion (Buddhist philosophy) | Morality | Law, regulations |
| **Others** | 1. Citizens were urged to sign pledge cards 2. The US Food Administration was responsible | 1. Various targeted models: home model, school model, military model, restaurant model 2. The pledgers (except for children) were obligated to donate 1 dollar | 1. The campaign is advocacy, not compulsion 2. IN-33 offline activities were mainly in Beijing | 1. Backed by anti-food waste laws 2. The target group extends from officials to the public |

Although the movements in South Korea and China (CYPC I) are led by social organizations, Ecobuddha was established early (March 1988) with a good foundation, while IN-33 was only an ad hoc team whose members all had their own jobs. In addition, the South Korean movement has developed a variety of targeted models for practice. By contrast, the Chinese movement has not proposed specific programs and lacks unified leadership and direction in its response activities. Moreover, the Clean Plate movement in South Korea draws on religion. In China, CYPC has relied on personal virtue.

In addition to clean plate movements, there have been many movements and programs to tackle food waste. Specifically, some have been dedicated to raising awareness of food waste, such as the "Love Food, Hate Waste" campaign in the UK, "Too Good for the Bin" in Germany, the Zero Waste Movement in Portugal, etc. [25]. Others have focused on donating excess food, such as the Last Minute Market project in Italy, the "Buy one, get one free later" initiative in the UK, and a series of events held by Feeding the 5000 in EU countries. Others provide cooking courses, such as a training program organized by the Bruxelles Environment (IBGE) in 2009. It was found that cooking classes can lead to the adoption of sustainable food practices [26]. For more specific projects or movements, Zamri et al. [27] gathered many attempts to curb food waste in both developed and developing countries, although he did not examine clean plate campaigns.

## 3. The Factors That Influence Food Waste

A wealth of research has focused on developed countries, where the largest amount of food waste is generated at the household level [28,29]. Over-preparation and excessive purchasing are the most significant causes of food waste [30,31]. Other factors that influence food waste include planning, shopping routines [32,33], the intention to prevent food waste [34], lack of knowledge about food storage and handling [35], retail offers and promotions [36], purchasing food in large packages [37], interpreting food labels inaccurately [38], dining out [39], and rejecting suboptimal foods [40]. Socio-demographic factors such as household size and composition [32,41], income [42], age [43], sex [44], education level [45], and employment status [46] are also important predictors.

The relevant external social factors are behavioral interventions, cultural factors, political factors, and COVID-19.

### 3.1. Behavioral Interventions

Behavioral interventions gained the attention of policymakers around the world in 2010 [47,48]. In the field of food waste, one main type of behavioral intervention has been the dissemination of information, which has helped to change wasteful behavior [49]. However, the effectiveness of different messages may differ. Zhang et al. [50] stated that interventions intended to provide information are important, but what they say is even more important. For example, a simple prompt-style message (encouraging people not to waste food) is more effective than a message that simply states the amount of food waste [51]. Providing information about the negative effects of food waste significantly reduces food waste, whereas providing information about composting food waste generates more waste [52].

In terms of clean plate campaigns, relevant posters are available from China and the US, but not from Korea, by searching the Google and Baidu engines. As shown in Figure 1, the posters at the top are from the US Food Administration and the posters at the bottom are from the Chinese government and netizens (the specific websites (URLs) are listed in Supplemental Table S3).

As seen from these posters, the slogans of the US movement combined food with the idea of victory in war, which is related to civil rights and national honor. By contrast, the Chinese slogans focus more on personal morality and character, and cleaning the plate is always presented as a virtue or a character trait required for personal success. However, the binding force of morality is weak.

There is a lack of empirical studies comparing the effects of the two campaigns. Google Trends can show search interest for a specific topic or term from 2004 onwards. Given the different market shares held by Google in China and the US, this paper could not directly compare specific values of interest. As shown in Figure 2, although the American Clean Plate Club was originally launched in 1917, the public's interest in the topic has never faded. The geographical areas of interest are concentrated in Canada and the USA. Interest in China's Clean Your Plate Campaign peaked in January 2013 and August 2020, coinciding with Xi Jinping's two public statements about limiting food waste. This is consistent with

the results of the Baidu Index (the Baidu Index is the big data analysis platform of Baidu, which is currently the most popular search engine in China. Baidu index is similar to Google Trends) (Supplemental Figure S1). The Baidu Index shows that there were very few searches or reports at other times, indicating that attention to CYPC in China has been sporadic rather than continuous. Empirical studies have found that CYPC I has had limited effects [53,54]. There have not been enough studies on the effect of CYPC II.

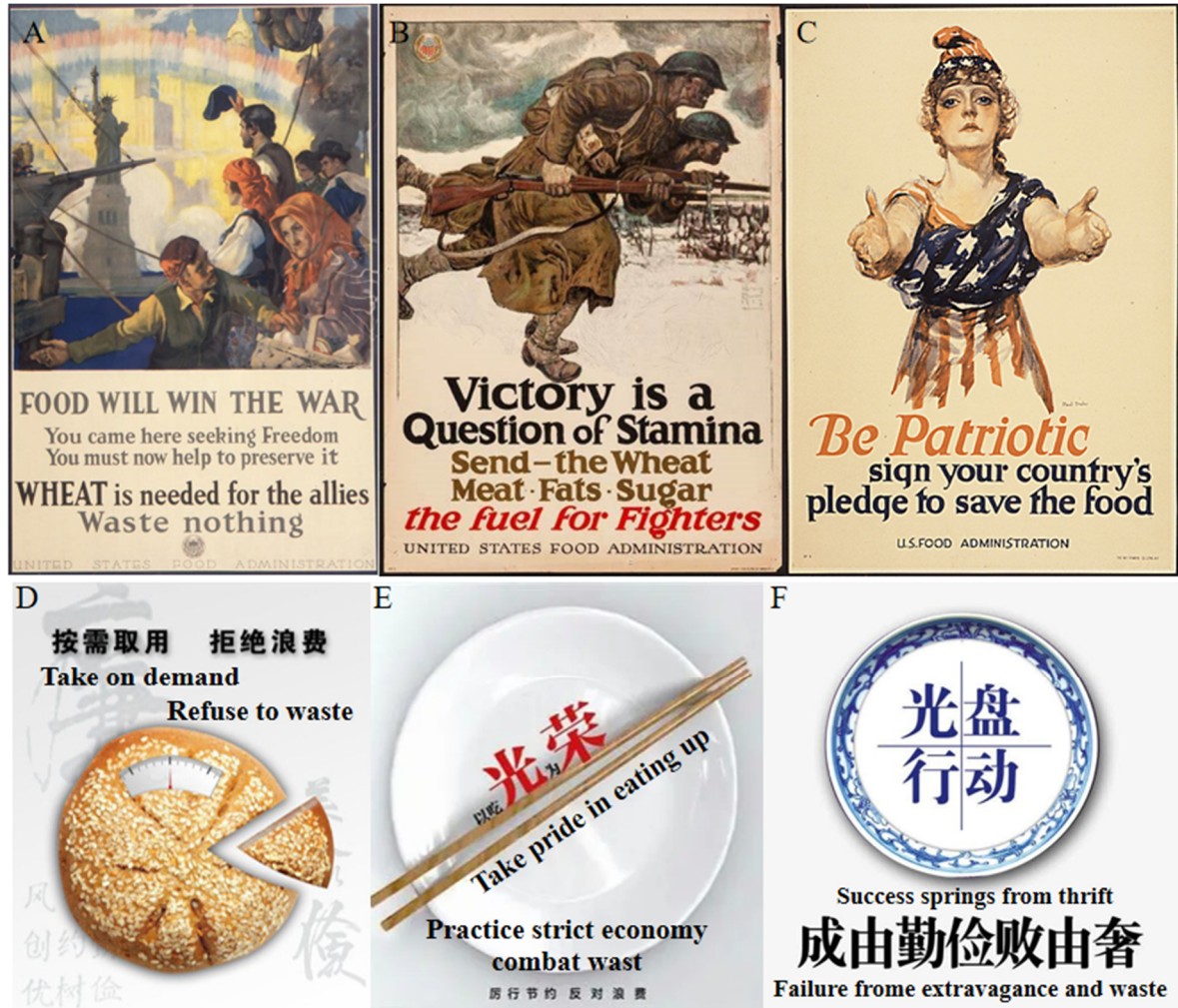

**Figure 1.** Posters related to clean plate campaigns from the United States and China, with translations. (**A**) Food will win the war; (**B**) Food is the fuel for fighters; (**C**) Saving food is patriotic; (**D**) Take on demand and refuse to waste; (**E**) Eating up food is glorious; (**F**) Waste leads to failure.

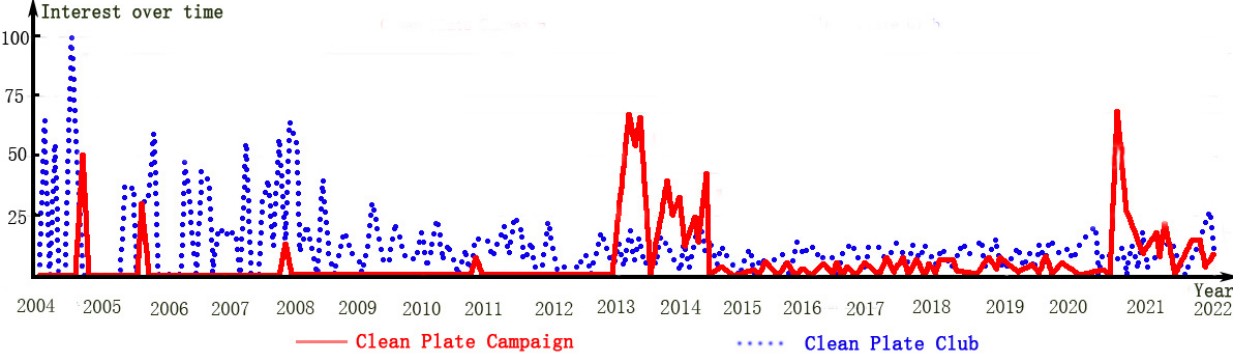

**Figure 2.** Interest over time in clean plate movements according to Google Trends.

In addition to information-based interventions, Metcalfe et al. [55] summarized other nudge interventions: placement/convenience and variety/portions. Adams et al. [56] discovered a relationship between the location of a salad bar and the waste of fruit and vegetables. Changing plate and portion sizes are also effective at limiting food waste. One experiment indicated that reducing plate size can reduce food waste by 19.5% [57]. Wansink and Van Ittersum [58] found that buffet diners with large plates wasted 135% more food than those with smaller plates. Furthermore, permanent (i.e., hard plastic) plates [59], the introduction of fridge cameras or food sharing apps [60], and interventions by Home Labs (a collaborative experiment with householders) [61] were also reported to be effective.

### 3.2. Cultural Factors

Diet composition, consumption norms, and the significance of food vary widely between countries and are culturally specific [62]. Food habits are determined by multiple stimuli pertaining to the dietary framework of a particular culture or religion [63].

Some scholars link religious beliefs to food waste [64]. Religion has a positive influence on sustainable consumption practices [65,66]. One study revealed the significant impact of religiosity on attitudes and subjective norms about food waste reduction [67]. Aschemann-Witzel et al. [68] indicated that religious beliefs are a critical factor when it comes to the degree of motivation to avoid food waste. In general, religions discourage food waste. In Christianity, one passage in the New Testament states: "When they had all had enough to eat, Jesus said to his disciplines, 'Gather the pieces that are left over. Let nothing be wasted'" (John 6:12 [NIV]). Similarly, Judaism has the Talmudic concept of *bal tashchit*, which can be interpreted as "thou shalt not waste" [69]. In the case of Islam, the Holy Quran states: "eat and drink, but don't waste. Indeed, He likes not the wasteful". In this religion, wasting food is a sin and an act of ingratitude; this perspective has proven to be one of the strongest drivers of reducing food waste [44]. However, it was discovered that restrictive religious norms (e.g., rules about food consumption and fasting) lead to greater food waste [70]. For example, Elmenofi et al. [71] found that food waste increases during the fasting month of Ramadan (an Islamic month of fasting) in Egypt. A review found that nearly 25–50% of the food prepared in some Arabic countries during Ramadan is wasted due to over-preparation [72].

However, Elshaer et al. [73] found that religiosity had a very weak negative influence on people's intention to waste food, whereas food consumption culture has a significant positive influence on people's intention to waste food. The characteristics of a nation's culture can affect the quantity of wasted food [74,75]. First, different cultures develop different attitudes towards food and food waste. Not all food waste is considered unacceptable in all cultures. Since meat is still at the center of most Danish dishes, it is more culturally acceptable in Denmark to throw away foods that are thought of as having lower status, like bread and vegetables [76]. Moreover, it is considered inappropriate to finish all the food on one's plate in Abu Dhabi [74,77]. In terms of moral judgments, the Masai think food wasting is more immoral than the Yali and the Poles [78]. Second, cultural contexts shape eating habits [15]. French traditions of moderation (versus American traditions of abundance) and a focus on quality (versus quantity) mean that the French eat less than Americans [79]. The Chinese dine together, and the dishes are often served in large portions. Conversely, Westerners tend to eat separately, and each person is assigned a limited amount of food on their plate. Third, cultural norms influence portion sizes [80,81]. China is influenced by collectivism. Each dish is served in a large portion to ensure that everyone has enough. By comparison, Japanese food is served in small portions. Japanese culture contains the concept of *mottainai*, which expresses the regret of wasting something valuable [82].

Two cultural values, namely hospitality and face-saving (in Chinese, *mianzi*), play an important role in food waste across many cultures. In terms of hospitality, Saudis and Arabs place a high value on generous hospitality. They provide guests with large quantities of food to exceed their expectations [73,83]. Slovenians and Chinese people also offer guests more food than is necessary [84,85]. Elshaer, Sobaih, Alyahya, and Abu Elnasr [73] have

noted that this culture of serving a lot of food to show hospitality means that people are highly likely to waste food. There is also the concept of being a "good provider", which means that one shows hospitality to family and friends by providing lots of food. Various scholars argue that the "good provider" mentality is a barrier to tackling the problem of food waste [85,86]. Wang, McCarthy, and Kapetanaki [85] have stated that expressions of hospitality through food are central to the social fabric of most societies and that "good provider" norms, along with socio-cultural factors, help to explain food waste behavior.

The American preacher Smith [87] has stated that face-saving is the most basic and important characteristic of the Chinese people. Face-saving significantly influences food waste [88,89]. In China, the concept of "face" is a highly valued quality related to reputation and ego. It is gained through success or boasting [90]. Specifically, preparing or ordering excessive meals is considered a sign of affluence or status, which allows people to "gain face". Doing the opposite causes people to "lose face". Moreover, China is a society based on favor (in Chinese, *renqing*), "favor" often refers to resources (tangible goods such as money and food, or intangible things such as help and opportunities) that are exchanged during interactions. This involves exchanges in a society of acquaintances, relying on moral constraints rather than regulatory constraints [91]. Treating people to dinner is the most basic way to exchange benefits, whether one is asking for a favor or thanking someone for help. Over-ordering is a common way of showing sincerity. Wang et al. [92] argue that when food is considered to be a social tool rather than a delicacy or necessity, food waste becomes an issue of low concern.

### 3.3. Political Factors

Culliford and Bradbury [93] found that consumers in the UK remain resistant to some aspects of sustainable diets. They highlighted the fact that policy action is required to enable behavioral change. Many countries have implemented policies or programs to limit food waste. According to a report produced in 2011, more than a hundred food waste prevention initiatives have been launched in EU countries [94]. Policy factors have been shown to curb food waste to a large extent [95,96]. Revised meal standards and policies have been found to significantly lower plate waste in school cafeterias [97]. Secondi, Principato, and Laureti [12] have emphasized the need for public–private partnerships and diverse policy interventions at the local level (i.e., targeting community-based interventions).

In China, the requirement to improve laws and regulations has appeared eight times in the previous 26 policies (Supplemental Tables S1 and S2). It was first mentioned in the 1996 policy. It was not until 29 April 2021, that China's first official anti-food waste law was enacted. Other countries have also enacted laws against food waste, such as France, Italy, and Japan. These countries are compared in Table 2. For its categories of analysis, this study referred to Vittuari [98], adopting the approach, instruments, and stages used in that study. Policy approaches to food waste prevention can be classified as suasive (e.g., communication campaigns, educational activities) or regulatory (e.g., legal obligations to donate surplus food, mandatory standards). Instruments can be market-based (e.g., subsidies, taxes, and tax concessions) or public services (e.g., food donation infrastructures) [98]. Food waste prevention pyramids, such as disposal, recovery, recycling, reuse, and prevention (ranked from least to most effective) have also been introduced as ways of studying the effectiveness of anti-food waste laws and policies [99].

France requires large stores to sign agreements with charities or face hefty fines. "The law is wrong in both target and intent," argued Jacques Creyssel, the head of a distribution organization for big supermarkets. "Big stores are already the pre-eminent food donors" [100]. Meanwhile, charities face the challenge of managing large quantities of food which are at times low-quality or inedible [101]. With extremely limited storage and distribution capacity, they must often discard uneaten food and bear the disposal costs [102]. Furthermore, Eubanks [103] argued that public awareness campaigns and the resulting cultural changes would more effectively eliminate food waste than fines and penalties. France's 2016 law focused on the retail sector, while the 2019 law extended to the mass catering and food industry.

**Table 2.** Anti-food waste laws in different countries.

| | France (https://www.legifrance.gouv.fr/jorf/id/JORFTEXT000032036289, Accessed on 16 March 2022) | Italy (https://www.gazzettaufficiale.it/eli/id/2016/08/30/16G00179/sg, Accessed on 16 March 2022) | Japan (https://perma.cc/LK8H-A8KU, Accessed on 16 March 2022) | China (http://www.npc.gov.cn/npc/c30834/202104/83b2946e514b449ba313eb4f508c6f29.shtml, Accessed on 16 March 2022) |
|---|---|---|---|---|
| **Adoption time** | 11 February 2016 | 19 August 2016 | 24 May 2019 | 29 April 2021 |
| **Main approach** | Regulatory | Suasive | Suasive | Suasive |
| **Main instrument** | Public services | Market-based | Public services | Public services |
| **Main objects** | Retailers, charities | General public, catering service providers | Governments | Governments, catering service providers |
| **Main stage** | Re-use, prevention | Prevention, re-use | Prevention | Prevention |
| **Funding** | No | Yes | No | No |
| **Donations** | Mandatory (Fine) | Incentive (Tax reduction) | Supported | Guided |
| **Main points** | 1. Food retailers are forbidden to destroy unsold food products still fit for consumption. 2. Obligation to establish a partnership with a charity organization to donate unsold food products, for stores over 4305 square feet. | 1. Municipalities may apply a waste tax reduction for entities engaged in food donation. 2. For a donation below €15,000, no official procedures are required. 3. Donation of products that are beyond the minimum term of conservation is possible 4. The law establishes a stakeholders' committee on food waste. | 1. Local municipalities will be urged to draft and act on their own plans. 2. October becomes the annual Food Loss Reduction Month 3. The law also sets up a body for the promotion of food loss reduction within the Cabinet Office. | 1. An anti-food waste supervision and inspection mechanism should be established. 2. Restaurants must provide small portions, display anti-waste signs, and offer packing services. 3. Prohibit the spread of content promoting food waste (overeating) or face fines. |

Unlike the French law, the Italian law has instead focused on incentives. Moreover, there are five provisions of funds in the Italian law to provide support, while no specific funding is given for measures against food waste in the French law [99,104]. Busetti [105] summarized the innovative measures of the Italian law, namely the possibility of donating food after the best-before date and a significant simplification of the bureaucracy of donations. At the same time, they saw a number of constraints, such as failure to address donors' propensity to donate, the capacity and will of charities, and the reputational risk implied by food after the best-before date [105].

Although the Japanese law specifies that it is the responsibility of both the national government and local authorities to reduce food loss, it lacks specific requirements for measures from companies or precise plans from authorities to enact such steps [106]. Japanese law thus acts as more of a rough guide. In addition to regulating governments, Chinese law gives more detailed requirements for other targets, especially catering service providers. Moreover, Chinese law also lists penalties for non-compliance but does not give objective and operational criteria for judging. Food donations are also briefly mentioned in the Chinese and Japanese laws, but these regulations are not as important or actionable as in France and Italy. Ayako [106] proposed that ambiguities related to responsibility for issues such as hygiene when providing food to charities have hindered such donations in Japan.

*3.4. COVID-19 Factors*

Following the outbreak of the COVID-19 pandemic, food production and consumption systems have undergone significant changes [107]. Food consumption routines and the perspective of consumers on food have also changed [108]. Specifically, lockdown and quarantine policies have forced some dietary and routine changes, such as increases in online shopping, less eating out [109], and increases in people's culinary skills [110,111]. Researchers have also discovered a shift towards healthier diets (e.g., increasing the intake of fruits and vegetables) [112] or unhealthier diets (e.g., eating more processed foods) [113]. Other changes include an increase in the consumption of local or domestic products [111,114], as well as weakened purchasing power (due to unemployment or pay reduction) [110].

Many scholars have studied the impact of the pandemic on food waste. The pandemic has led to stockpiling and panic buying [115,116]. Non-perishable food items were prioritized by stockpilers [110]. For example, Ben Hassen et al. [117] have shown the extent of panic purchasing in Lebanon, with 73.13 percent of respondents reporting that they stocked up on food once COVID-19 became serious. The same phenomenon was observed in Morocco [118]. People's reasons for stockpiling were the fear of food shortages and price rises, an attempt to reduce store visits to limit exposure to COVID-19 [1,119], and a need to feel in control of a perilous situation [120]. Cosgrove et al. [121] found that food stockpiling was a predictor of increased overall food waste. Recent data show that individuals did not consume a good proportion of their stockpiled food, leading to increased waste [122,123]. Other reasons for increased food waste include unappetizing taste and food exceeding the expiration date or rotting [124].

However, most studies show that the pandemic has led to a reduction in food waste and a shift towards more sustainable patterns of consumption [110,125,126]. These studies mostly took the form of online questionnaires or self-monitored reports. Based on data derived from the waste management department in Malaysia, one study estimated that there was a 15.1% decrease in food waste during the lockdown in the country [127]. There are many possible reasons for this decrease in food waste. First, COVID-19 improved people's awareness about food and food waste [114,128,129]. Second, the COVID-19 lockdown improved people's ability to manage food, encouraging them to plan their food shopping more effectively, enhance their in-house food storage, and use up more leftovers [120,130]. Third, Pappalardo et al. [131] found that concerns about the impact

that the pandemic could have on the waste management system and the desire not to add pressure to that system both helped to decrease food waste.

Overall, the impact of COVID-19 on food waste has varied from country to country. This is due to epidemiological circumstances, the baseline socio-economic situation, and the shock resilience of different countries [116]. Rodgers et al. [132] found that people in the US reduced their relative food waste to a greater extent than people from Italy. However, the effects have also varied from person to person. Amicarelli et al. [133] identified three different clusters after the COVID-19 lockdown: some people wasted more food and some wasted less. Armstrong et al. [134] found that participants with food security wasted a smaller percentage of purchased and cooked foods compared to participants without food security. Other influencing factors included age, gender, household composition, and employment status [135–138].

## 4. Discussion

The clean plate campaigns in different countries have focused on different methods. Specifically, the Clean Your Plate Campaign in China relies heavily on personal ethics, with a short-term public attention span. By comparison, the "Clean Plate Club" in the US was a continuous government program in which cleaning plates was defined as a patriotic duty, with long-term effects. In American adults, approximately 92% of self-served food is eaten [17]. The Clean Plate movement in South Korea is based on religion and is highly operational. In short, political power and religion can be better tools than morality for a clean plate movement. However, Stoddard [139] believes that obesity in the US today is an echo of the Clean Plate Club of yesteryear. Velez, Majda, and Wansink [16] also found a positive correlation between parents who were told to clean their plates as children and their current BMI, which also appeared in the next generation. The implementation of cleaning plates should, therefore, be more flexible. Furthermore, the government and social organizations have assumed different roles in the clean plate movements in the three countries. Many existing anti-waste movements and programs were initiated by NGOs with government funding. Public–private partnerships are important as well [12].

Cultural contexts shape eating habits; however, food culture itself and its influences are subject to change. For example, in ancient China, people used to eat separately. At that time, there was a strict hierarchy between emperor and subjects, fathers and sons. Accordingly, people at different levels of status ate different foods and thus had to dine separately. With the collapse of the feudal hierarchy, the integration of ethnic groups, and people's greater enthusiasm for banquets, dining separately began to turn into dining together in the time of the Northern and Southern Dynasties (420–589) [140]. Moreover, other national cultures, brought by colonization or globalization, can also have an impact on native cultures. In Brazil, an abundance of food on the table has been prevalent since the colonial period, influenced by Portuguese culture [141]. Aleshaiwi and Harries [142] found that Saudi culture has changed rapidly since the 1980s when the opportunity for local people to interact with other cultures increased. Furthermore, increases in income may diminish the impact of an anti-waste culture. A review has suggested that rising disposable income is one of the four drivers of food waste [8]. With more disposable income, people eat not just for the sake of satiety but also to pursue enjoyment. A study has found that consumers consider waste to be acceptable when they aim for taste and convenience [143].

Many countries have resorted to laws to tackle food waste. French law bans supermarket waste and requires food donations. It has been suggested that if the United States were to model law on the French statute, it would be criticized as an overreach of state power [103]. Legislation must take into account the special circumstances of different countries. This study also found that the French and Italian laws (Table 2) pay great attention to reuse at the retail stage as the main way to reduce food waste, especially in the French law. However, retail only accounts for a very small proportion of total food waste [144]. More focused and targeted laws might be more effective. All four laws in Table 2 address food waste prevention, including awareness and education campaigns. A survey shows that

people waste 10 times more food than they think they do [145]. More publicity for food waste is therefore needed in the future.

The COVID-19 pandemic has shaped food waste prevention behaviors, such as cooking at home, noticing food waste, and using up leftovers. Several scholars have stated that COVID-19 might have opened a "window of opportunity" for encouraging a shift towards more sustainable food consumption habits and lifestyles [114,116]. However, it is worth noting that consumer changes might have been driven more by the socioeconomic factors of the COVID-19 lockdown (i.e., food availability, restricted movements, loss of income), than by pro-environmental concerns [130,146]. In other words, when the COVID-19 pandemic has passed, wasteful behaviors may return as before. After all, there have been many other pandemics in the past. Roe et al. [147] believe that although COVID-19 led to a reduction in household waste, pandemic-driven disruptions may induce larger intermittent purges of food due to changes in work patterns, food services, and food retail. In the future, it will be important to continue and reinforce the good habits formed during the pandemic.

## 5. Conclusions

This study has examined clean plate campaigns and analyzed the factors influencing food waste to draw retrospective lessons for food conservation. The movements examined have varied from country to country. The Clean Plate Club in the US took place in wartime and relied heavily on political power to ensure compliance. The Clean Plate movement in South Korea was based on religion and targeted implementation models for different groups. The CYPC in China relied more on personal virtue and lacked incentives.

To improve the effectiveness of the campaigns, the joint participation of the government, social organizations, and the public is needed. The government should give attention and provide support at the national level, such as funding, laws, regulations, and specialized institutions. Moreover, social organizations should provide specific and detailed implementation programs and carry out activities regularly. If public consciousness is weak, appropriate incentives are necessary in the context of long-term school education and media publicity. In general, the fundamental aim of all the above measures should be to raise awareness of conservation and to shape a culture of food conservation. In the age of social media, social media and celebrities can also be used to promote and shape the cultural fashion for food conservation.

Food waste is the result of multiple factors. Religion and cultural values, such as hospitality and face-saving (*mianzi*), are factors that influence food waste. When it comes to anti-food waste laws, China's law relies on persuasion and is not clearly enforceable. By contrast, laws in France and Italy focus on re-use and rely on coercion and incentives. COVID-19 has greatly affected people's consumption behaviors and habits, leading to a general reduction in food waste.

We have not conducted any empirical studies, so there are no data to compare the actual effects of each campaign and determine which is the most effective. This is a limitation of this study. It is recommended that quantitative studies be conducted to compare the implementation effects of various movements and measures among different countries. Moreover, different interventions may have different effects, so it is important to explore the most relevant and effective interventions for different scenarios.

**Supplementary Materials:** The following supporting information can be downloaded at: https://www.mdpi.com/article/10.3390/su14084699/s1, Figure S1. Baidu Index data for the keywords "Clean Your Plate Campaign"; Table S1. Names and URLs of policies related to food conservation in China (1990–2021); Table S2. Highlights of policies related to food conservation in China (1990–2021); Table S3. Sources of sample figures.

**Author Contributions:** Conceptualization, G.W.; investigation, L.W. and Y.Y.; writing—original draft preparation, L.W. and Y.Y.; writing—review and editing, G.W.; supervision, project administration and funding acquisition, G.W. All authors have read and agreed to the published version of the manuscript.

**Funding:** The research was supported by the Key Project of the National Social Science Fund of China (grants 20FXWA003).

**Institutional Review Board Statement:** This study did not involve human or animal studies.

**Informed Consent Statement:** This study did not involve human studies.

**Data Availability Statement:** This study did not report any data.

**Conflicts of Interest:** The authors declare no conflict of interest.

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
