# Peer review of "The Clean Your Plate Campaign: Resisting Table Food Waste in an Unstable World"

_sustainability, doi:10.3390/su14084699_

Round 1
Reviewer 1 Report
I have read carefully the paper and find it perceptive and important for the scientific community. However, certain issues must be answered. In my view, the manuscript needs to be improved.
My suggestions are listed below:
line 94 - remove the reference in this form, put the following number in brackets
Table 1 and Table 2 - please, reorganize tables in order to make more simple to read them
line 165-168 provide this information in the text where you mentioned Figure 1, not as "notes"
Figure 2 - please add X and Y axis names
Conclusions - add potential steps for improvement and solutions for this campaign.
Author Response
|
1. line 94 - remove the reference in this form, put the following number in brackets |
|
Response: Thank you for your reminder. We have changed this and re-updated all the references through EndNote. |
|
Table 1 and Table 2 - please, reorganize tables in order to make more simple to read them |
|
Response: Thank you for your kind advice. We changed the alignment of the text to be horizontally centered. |
|
3.line 165-168 provide this information in the text where you mentioned Figure 1, not as "notes" |
|
Response: Thank you for your advice. We have consolidated the notes into the first paragraph below Figure 1. |
|
4. Figure 2 - please add X and Y axis names |
|
Response: Thank you for your reminder. We have added the labels; please confirm that the new image is acceptable. |
|
5.Conclusions - add potential steps for improvement and solutions for this campaign |
|
Response: Thank you for your pertinent advice. We have added potential steps according to your helpful suggestion in the second paragraph of the conclusion section. |

Reviewer 2 Report
I have read the previous version of the article, and it has significantly improved since then.
Author Response
|
1. I have read the previous version of the article, and it has significantly improved since then |
|
Response: Thank you for your affirmation and recognition. Also thank you very much for your valuable comments in the last round of reviews. |

Reviewer 3 Report
This is a very interesting and very well written review on the food waste campaigns in the world. Generally, the paper is well structured and easy to learn. Some observations are made below.
L94: Please see citation style.
L96: This sentence is repeated many time in the manuscript. Please reduce repetition.
Tables and figures should be mentioned before in the text, not after. Please revise in the whole manuscript.
Table 1, 2: Please revise text alignment because it is confusing where the row ends and starts the next one.
Figure 2: The artwork is bad: what is "Note" mention on the figure? The legend lacks, what the colors mean? The size is not proportional.
Author Response
|
1. L94: Please see citation style. |
|
Response: Thank you for discovering this omission. We have changed this and re-updated all the references through EndNote. |
|
2. L96: This sentence is repeated many time in the manuscript. Please reduce repetition |
|
Response: Thank you for your reminder. We have deleted the sentence in line 96. Currently, a similar sentence only appears in the abstract. |
|
3. Tables and figures should be mentioned before in the text, not after. Please revise in the whole manuscript |
|
Response: Thank you for your comments. We have repositioned Figure 2 and Table 2 so that they appear before the textual descriptions associated with them. The positions of Figure 1 and Table 1 are fine. |
|
4. Table 1, 2: Please revise text alignment because it is confusing where the row ends and starts the next one |
|
Response: Thank you for your suggestions. We changed the alignment of the text in the whole table to be horizontally centered. |
|
5. The artwork is bad: what is "Note" mention on the figure? The legend lacks, what the colors mean? The size is not proportional. |
|
Response: Thank you for your pertinent suggestions. We have added the legend and axis names. In the new Figure 2, we differentiate not only by color but also by dotted and solid lines. |

Round 2
Reviewer 1 Report
Dear Authors,
Thank you for the revised version of your manuscript. After the second revision, the paper has improved.
This manuscript is a resubmission of an earlier submission. The following is a list of the peer review reports and author responses from that submission.
Round 1
Reviewer 1 Report
Dear Editor, Dear Authors,
I have the opportunity to review an article entitled „The Clean Your Plate Campaign: a hard way to resist food waste“. Although I encourage the concept besides this article, I have to reject this article.
First, of all, line 1 suggests that the Authors did not sure this was a research or review article? And that is a very big problem because the Sustainability is the journal for scientific-relevant, actual topics. In this form, this paper is a marketing flyer and promotion for some campaign.
If the Authors decide to make a scientific-relevant paper, with a statistical approach framed with the topic of the Clean Your Plate Campaign, they need to represent the results of many respondents to have a relevant examination. China is a big country, and if China is a region of interest for this campaign, the Authors need to have a sufficient number of samples. Also, the concept of displaying this type of research is very sensitive, so the Authors need to be very systematic, concise, with a clear presentation of the obtained results.
Besides the wrong direction of this paper, the Authors don't have affiliation, e-mails, corresponding authors?
Furthermore, the Authors did not read the Guideline for the preparation of the manuscript. All references are incorrect in the text, but also Reference List is not following the Guideline.
I would not comment on everything else at this stage of the work.
Overall, this paper is not good enough for publishing in the Sustainability. I support resubmitting this paper once the conceptual issues the paper currently has are resolved.
Reviewer 2 Report
An interesting literature review is this paperwork. It is presented that the COVID-19 pandemic further threatens world food security and has created an urgent need for food conservation, fact accepeted by all the actors involved. This reasearch coud be more extended, more detailed and more specific, but on the whole, is a good paperwork.
Reviewer 3 Report
The article presents a timely and interesting research about various "clean your plate" movements, but the main focus is on the Chinese one.
A thorough investigation was conducted by reviewing the appropriate literature as there are many references that can be found in the article. It should be noted that the format of the references as well as of the citations ([1], [2,3], [4-6]) should be changed to the MDPI journal format.
Even though a thorough investigation was conducted, there is no consensus on which movement is the optimal one. Based on the results of the investigation, which one could produce the least waste? Or, based on their pros and cons, can a more effective, new movement be created from their characteristics?
Also, the reviewer suggests that the type of the paper should be changed to "Review" (as in literature review) instead of "Article" in the very beginning. Based on the contents of this paper, it is more fitting (at this moment).